# Detection of Organic Substances by a SERS Method Using a Special Ag-Poly(Chloro-P-Xylylene)-Ag Sandwich Substrate

**Irina Boginskaya** [1], **Aliia Gainutdinova** [2,\*], **Alexey Gusev** [1,\*], **Karen Mailyan** [1], **Anton Mikhailitsyn** [1,\*], **Marina Sedova** [1], **Artem Vdovichenko** [3,4] , **Ilya Ryzhikov** [1,5], **Sergei Chvalun** [3,4] and **Andrey Lagarkov** [1]

1. Institute for Theoretical and Applied Electromagnetics Russian Academy of Sciences, 125412 Moscow, Russia; i.boginskaya@bk.ru (I.B.); k.mailyan@gmail.com (K.M.); sedova_marina@mail.ru (M.S.); nanocom@yandex.ru (I.R.); itae@dol.ru (A.L.)
2. Moscow Institute of Physics and Technology (National Research University), Dolgoprudny, 141701 Moscow, Russia
3. National Research Center "Kurchatov Institute", 123182 Moscow, Russia; vdartem@ya.ru (A.V.); s-chvalun@yandex.ru (S.C.)
4. N.S. Enikolopov Institute of Synthetic Polymeric Materials RAS, 117393 Moscow, Russia
5. FMN Laboratory, Bauman Moscow State Technical University, 105005 Moscow, Russia
\* Correspondence: zombiephysic@gmail.com (A.G.); 4839356@mail.ru (A.G.); krak1111@gmail.com (A.M.)

**Abstract:** Spectroscopy based on surface enhanced Raman scattering (SERS) is widely used as a method with extremely high sensitivity for molecular and chemical analysis. We have developed thin-film sandwich structures, in which, when used as sensitive elements for detecting organic compounds at low concentrations, high-amplitude spectra of surface enhanced Raman scattering are observed. Using gas-phase cryochemical synthesis and thermal sputtering in vacuum, SERS active sandwich structures Ag–poly(chloro-p-xylylene)–Ag (Ag–PCPX–Ag) were obtained. In the process of creating sandwich structures, the upper silver film takes the form of a complex island topology with submicron sizes. A series of samples were made with different thicknesses of the polymer and upper silver layers. SERS spectra of the analyte chemically adsorbed on the film surface were obtained, demonstrating a significant amplification (up to $10^4$) compared with the control sample. The dependence of the gain on the silver concentration is characterized by a maximum polymer layer thickness of 600 nm and a 30 nm thick upper silver layer. A selective amplification of the low molecular weight compound spectra with respect to proteins was observed. A semi-empirical model is proposed that is in good agreement with the experimental results.

**Keywords:** Raman spectroscopy; nanostructured metal film; nanocomposite; surface spectroscopy; DTNB

## 1. Introduction

SERS (surface enhanced Raman spectroscopy) is a promising method for chemical and biochemical analysis, due to its extremely high sensitivity [1–3]. The development of substrates based on silver or gold, which are characterized by efficiency, stability, and economy, is especially relevant for research in the field of molecular biology [4–6]. An important task is also the creation of substrates demonstrating the selectivity of the SERS method for the detection of biological objects in low concentrations [3]. The most commonly used technologies for the synthesis of substrates for SERS are the chemical synthesis of nanoparticles, the deposition of coatings based on metals, and other methods [7,8]. Each of

the substrate types obtained is characterized by a number of both advantages and disadvantages, and by its own selectivity and stability [9–11]. In particular, substrates based on nanoparticles require careful stabilization in solution and control of their charge states [12,13]. Based on the previously developed technology of metal-polymer composite cryo-synthesis in vacuum, it is possible to obtain coatings with the desired optical properties due to the stabilization of nanoparticles in the bulk and surface layer of the polymer matrix, which are characterized by long-term stability of their properties [1,14].

At present, lattice structures are often developed as SERS-active substrates (e.g., [15]). They are formed for direct application to large scale wafer substrates. In work [15], gratings with a square two-dimensional array of silicon nanostructures obtained by interference lithography, with a gold coating, were created with high accuracy and homogeneity using large available silicon casting technologies, and provided a high signal amplification up to $10^7$. Such methods are technologically quite complicated and expensive, and require preliminary accurate theoretical modeling. They are subject to high quality requirements for the boundaries of nanostructure elements. Additionally, a roughness of the order of 1 nm can lead to the lack of realization of the SERS effect.

Today it is becoming more and more popular to develop various sandwich structures for use in SERS studies. For example, in the study [16], a graphene oxide film was placed between thin gold and silver films obtained by magnetron sputtering. SERS analysis at low laser intensity recorded additional sharp Raman modes, sometimes stronger than the typical band. They arise in hot spots and can be associated with localized vibrations of molecular groups associated with graphene oxide. In [17], a series of thin-film sandwich structures with bowl-shaped silver cavities (BSSCs) and silver nanoparticles for SERS-detection of biomolecules were fabricated by electro deposition. The results showed that both the lower BSSC thin film and the upper layers of nanosilver contribute to the final amplification of the SERS signal. This method detected FITC-labeled protein (fluoresce in isothiocyanate (FITC)) with a detection limit of 0.1 ng/mL, and also detected unlabeled proteins.

Sandwich structures for quantitative molecular detection are being actively developed [18]. In this experiment [18], the analyte molecule was placed between silver nanoparticles (silver nanoparticles (SNPs)) prepared by the Lee–Meisel method [19] and silver nanostructures obtained on porous anodic aluminum oxide (AAO) by electro deposition. These substrates demonstrated good stability and reproducibility with high detection sensitivity for rhodamine (R6G) and melamine. It has been shown that this high SERS sensitivity is achieved due to the strong amplification of the electromagnetic field by localized surface plasmon (LSP) between two silver nanostructures.

In [20], a sandwich structure with gold nanoparticles integrated into silver nanosized hexagonal matrices separated by monolayer graphene of atomic thickness was developed by electron beam lithography. The graphene interlayer acted as a supporting layer between the metal nanoparticles and the arrays, and it also acted as a space to create gaps at the nanometer scale, and finally as a protective layer, preventing oxidation and acting as a collector of molecules to absorb the analyte. Modeling has shown that such a system of nanoparticles of gold, graphene and hexagonal silver nanoholes forms a significant number of hot spots with a strong increase in the electric field. Therefore, the manufactured sandwich structure has an ultra-high sensitivity with a detection limit at the sub-picomolar level and good reproducibility.

It is also possible to create special stable and reproducible Ag–graphene–Au nanostructures as an efficient SERS substrate fabricated by depositing reduced graphene oxide (rGO) between plasmonic layers of silver dendrite and gold nanoparticles obtained by a simple electro deposition method [21]. Higher SERS activity of substrates was obtained due to the combined effect of chemical enhancement of rGO with a large surface area with strong absorbing ability to target molecules and the electromagnetic enhancement of plasmon silver and gold under illumination. The enhanced interlayer interaction facilitated the efficient transfer of plasmon containing Ag–rGO surfaces from the silver layer to the upper gold layer using the rGO interlayer. In addition, it was shown that a decrease in the rGO intermediate layer affects the growth of gold on the rGO surface and the Raman signal for rhodamine B

molecules absorbed on a multilayer nanostructure, while demonstrating an 8-fold increase in the SERS RhB signal as compared to primary silver dendrites. A low detection limit of about 10 nM was shown.

In a study [22], composite thin Au–graphene oxide–Ag sandwich structures exhibited broader optical absorption and increased absorption intensity compared to a thin silver film. It was shown that the Raman signal for rhodamine B molecules on the basis of these substrates was enhanced by a bimetallic layer and a graphene oxide layer with controlled absorption intensity and fluorescence quenching effects.

The problem of reliable placement of an individual molecule of interest in a hot spot remains urgent in order to ensure its analysis at the level of one molecule for studying complex chemical and biological processes that cannot be easily investigated using ensemble methods. One study [23] described a method for locating and protecting one target analyte at a SERS hot spot in a plasmon nanojunction. A smart hotspot was created using thiol-functionalized cucurbit [6] uril (CB [6]) as a molecular spacer that binds the silver nanoparticle to a metal substrate. Experiments in this work have shown that it is possible to reproducibly treat the SERS substrate so that 96% of the hot spots contain one analyte molecule. In addition, it was found that the enhancement of the SERS signal decreases by one per square distance from the center of the hot spot, while the cross section of single SERS molecules increases with an increase in the diameter of the silver nanoparticle.

In the work presented by us, we propose to use the technology of cryochemical synthesis, which will allow one, in one process, to create a dielectric spacer based on PCPX that provides the dispersion of silver in the form of nanoparticles aggregates. This will make it possible to create a layer based on silver and polymer, characterized by plasmonic properties, with the self-assembly method, without involving additional techniques in one installation and without the use of any solvents and other auxiliary substances. The plasmonic properties of the coating should be verified using the ellipsometry method. In addition, such coatings are conformal and the lower layer of silver in the sandwich structure will provide adhesion and optical activity of the spacer, which should be verified by ellipsometry. To create such structures, polymer of the poly-(paraxylylene) groups was chosen, since they are unique—they polymerize from the gas phase without additional accompanying of organic substances [1,14]. This allows them to be deposited simultaneously with the flow of atomic silver nanoparticles. Of this group of polymers, poly(chloro-p-xylylene) is the most thermally and chemically stable. The proposed technological scheme allows simultaneous implementation of several requirements for SERS structures. The polymer layer provides optical activity for the entire structure and simultaneously forms a nanostructure based on dispersed silver in the form of aggregates of plasmonic nanoparticles. Such a geometrical scheme of the optical structure was first proposed for implementation in SERS applications. Coatings can be applied to any surface. We have proposed a polycor. However, any substrate that adheres to silver can be chosen, including surfaces of the high orders.

Thus, using cryochemical synthesis technology and thermal sputtering in vacuum, it was possible to reproducibly produce SERS-active sandwich structures of Ag–poly(chloro-p-xylylene)–Ag (Ag–PCPX–Ag). In the process of creating sandwich structures, the upper silver film was formed in the form of a complex island topology with submicron sizes depending on the deposition conditions, layer thicknesses, and morphology of the polymer base. The lower continuous layer of silver had good reflective properties. A series of samples was made with polymer layer thicknesses 300–2000 nm, with non-uniform silver coatings conforming to their surface with 10–65 nm thicknesses.

The substrates obtained were studied by atomic force microscopy (AFM), ellipsometry, and Raman spectroscopy. The enhanced properties of the substrates were analyzed based on the Raman spectra obtained of 5,5-dithio-bis-(2-nitrobenzoic acid) (DTNB) chemically adsorbed on the film surface. A significant enhancement of the spectrum was demonstrated (up to $10^4$) compared with the control sample [1]. The dependence of the gain on the sandwich's structural parameters is characterized by a maximum with a polymer layer 600 nm thick and silver layer 30 nm thick. The selective amplification of the spectral amplitudes of the low molecular weight compound with respect to proteins was observed.

## 2. Materials and Methods

### 2.1. Obtaining the Ag–PCPX–Ag Sandwich Structure

The polymer was precipitated in equipment specially designed and manufactured by our group [14] on polycrystalline $Al_2O_3$ (polycor) substrates cooled with liquid nitrogen with a silver layer 100 nm thick preliminarily formed on them from a Knudsen effusion cell. Using the same equipment, the Ag–PCPX–Ag sandwich structures were prepared by low temperature vapor deposition polymerization (VDP) technology [24]. The first stage is the co-condensation of chloro-p-xylylene (CPX) monomer on a silver layer prepared earlier, and after that, Ag vapor on the monomer layer cooled to −196°C. At such a low temperature, the monomer is metastable [25,26]. The CPX monomer was prepared by pyrolysis of a cyclic dimer of CPX at 670 °C using the standard Gorham's method [27]. Ag vapor was produced by thermal evaporation by the Knudsen effusion cell. The deposition rate of the co-condensate was approximately 10 nm/min. During the deposition, a residual vapor pressure below $10^{-5}$ Torr was maintained in the VDP chamber.

Upon the second phase of the process, with slow heating of the co-condensate up to room temperature, the polymerization of CPX into PCPX and aggregation of the Ag clusters and atoms into nanoparticles occurred on the top of the sandwich.

As a result, sandwich structure samples on a polycor plates were obtained, composed of a silver reflecting layer with a constant thickness of 100 nm ($h_3$), a PCPX layer with a thickness $h_1$, and a silver layer with a thickness $h_2$. The structure scheme is shown in Figure 1a. Table 1 presents the parameters of the manufactured samples. Figure 1b shows the AFM image of the $Al_2O_3$ amorphous polycor substrate surface used as the basis for the SERS structure creation.

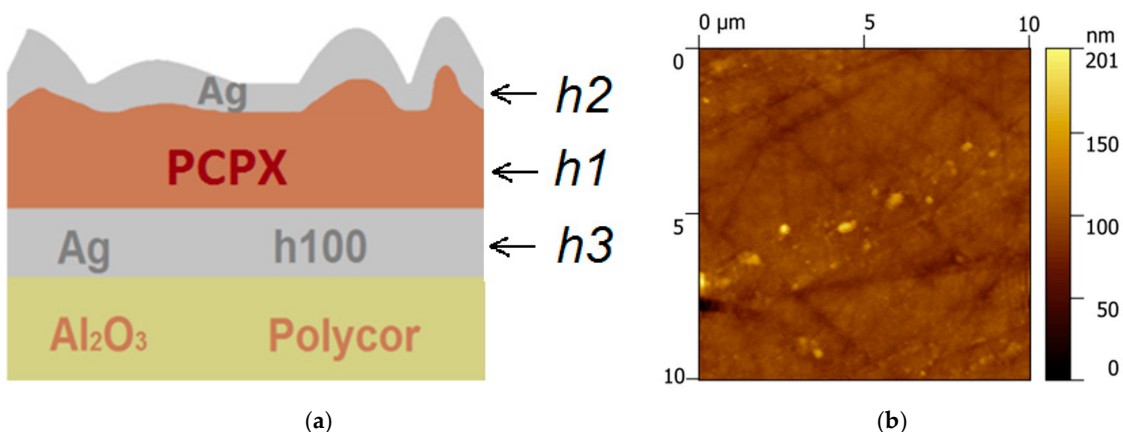

(a)          (b)

**Figure 1.** SERS structure based on sanswich Ag–PCPX–Ag: (**a**) Ag–PCPX–Ag structure schematic; (**b**) AFM image of polycore.

**Table 1.** Sample parameters.

| Polymer Thickness, $h_1$, nm | 300 | 300 | 300 | 600 | 900 | 900 | 900 | 1500 | 1500 | 2000 |
|---|---|---|---|---|---|---|---|---|---|---|
| Upper Silver Layer Thickness, $h_2$, nm | 10 | 20 | 30 | 30 | 30 | 50 | 65 | 20 | 30 | 30 |

Analysis of the AFM image of the polycor surface showed the surface characteristic of an amorphous material. The root-mean-square roughness of the polycor surface, according to the analysis of AFM data, does not exceed 6 nm.

## 2.2. Substrate Modification

To carry out SERS studies, the substrates were modified using a DTNB solution (≥98%, Sigma Aldrich, St. Louis, MO, USA) with concentration 0.2 g/L in ethanol (≥99.8%, Sigma Aldrich, USA). The substrates were immersed in the solution for 1 h. After extraction from the solution, they were immediately washed three times with ethanol (150 μL per wash) and dried with a stream of air. Next, SERS spectra were taken from all samples.

## 2.3. SERS Spectra Measurement

The SERS spectra were measured using an Alpha 300R Raman spectrometer (WITec, Ulm, Germany) based on a confocal microscope using a Zeiss Epiplan Neofluar 10×/0.25 lens (Oberkochen, Germany). To excite the spectra, a Toptica laser with a wavelength of $\lambda = 785$ nm was used. A laser power $P = 0.14$ mW was chosen. The spectra were measured at 20 random points of the sample. The accumulation time of one spectrum was 10 s. The spectra were processed in the OPUS 7.0 program (Bruker, Oberkochen, Germany). The baseline was adjusted using the rubberband correction algorithm.

Additionally, mapping of the SERS signal from samples over an area of $150 \times 150$ μm$^2$ with a resolution of $75 \times 75$ points was carried out. The scan time per point was 22.3 s. The area in the center of the sample was selected for scanning, and five maps were scanned for each concentration. After scanning, maps of the DTNB 1338 cm$^{-1}$ band amplitude distribution were constructed.

## 2.4. Morphology Investigation

Morphology studies were carried out using a Solver atomic force microscope (NT-MDT, Moscow, Russia) in a tapping mode. Areas with a size of $10 \times 10$ μm$^2$ were measured and surface roughness parameters were calculated using the Gwyddion software package (CMI, Jihlava, Czech Republic) using the built-in algorithm for determining the roughness parameters.

Parameters such as root-mean-square roughness $rms$, height over three points $R_{3z}$, kurtosis $R_{ku}$, root-mean-square waviness $W_q$, and average profile wavelength $\lambda_a$ were determined. A direct correlation was obtained between the SERS spectra amplitude and the sample morphological characteristics.

## 2.5. Investigation of Optical Properties

It is known that the appearance of SERS in metallic semicontinuous films requires the presence of plasmon resonance [28,29]. We investigated the dependence of the enhancing properties of the sandwich structures on their optical characteristics, which, in turn, strongly depend on the geometric parameters of the layer structure [24]. For all samples, the spectra of ellipsometric angles $\Psi$ and $\Delta$ were measured using a SAG 1891 ellipsometer (IFP SD SA, Novosibirsk, Russia) at incidence angles of 60 and 70 degrees in the spectral range 380–980 nm. To model the optical parameters of the films in the Spectr software (IFP SD SA, Novosibirsk, Russia), the Maxwell–Garnett effective medium model [30] was used, which makes it possible to represent an inhomogeneous film in the form of an equivalent homogeneous medium with an effective dielectric constant. As a result, it was found that only samples with polymer thicknesses of 300 and 600 nm and a silver layer of 30 nm were characterized by the presence of plasmon resonance near the exciting frequency.

## 3. Results and Discussion

### 3.1. Morphology Investigation

The results of the morphology study are presented in Figure 2. It can be seen that all the samples have a complex surface structure, depending on the ratio of the thicknesses of the polymer and silver layers. As mentioned above, during the formation of the sandwich structure, a layer of chloro-*p*-xylylene monomer was first formed, and then a silver layer was applied on top of it.

With subsequent monomer polymerization, the surface morphology of the polymer layer changes significantly. Silver on its surface also takes the form of various structures. At polymer thicknesses of up to 300 nm, agglomerates up to 400 nm in size are formed on the surface, consisting of silver nanoclusters with sizes 40–80 nm, as shown in Figure 2a,c. When the polymer thickness is doubled, as shown in Figure 2d, the surface looks like a nanoisland film of silver monoclusters. With a further increase in the polymer thickness, the surface is a submicron islet nanostructured silver film with a surface filling coefficient that grows with increasing silver thickness, as shown in Figure 2e–j.

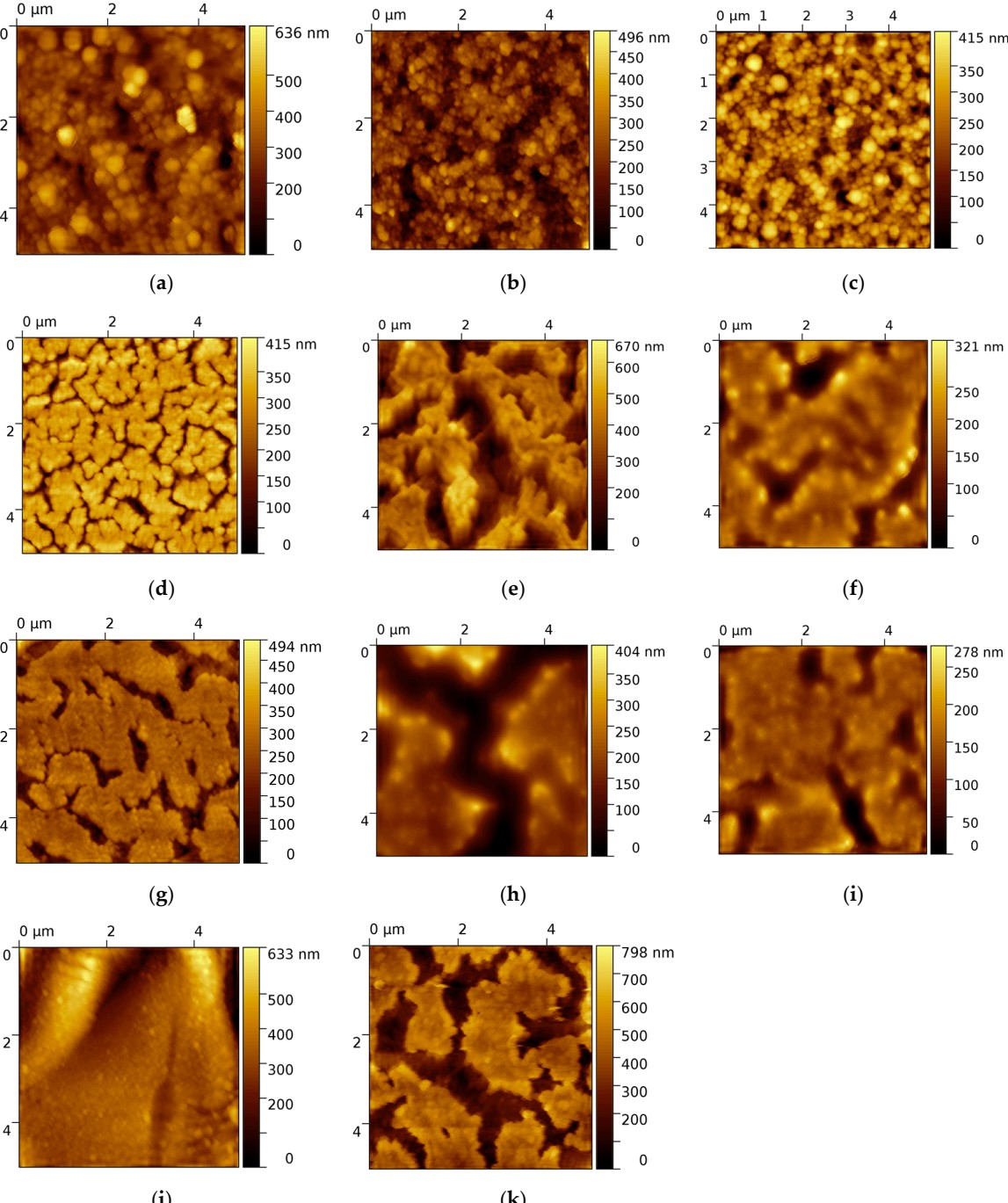

**Figure 2.** AFM surface, $h_1$—PCPX thickness, $h_2$—Ag thickness: (**a**) $h_1$—300 nm, $h_2$—10 nm; (**b**) $h_1$—300 nm, $h_2$—20 nm; (**c**) $h_1$—300 nm, $h_2$—30 nm; (**d**) $h_1$—600 nm, $h_2$—30 nm; (**e**) $h_1$—900 nm, $h_2$—30 nm; (**f**) $h_1$—900 nm, $h_2$—50 nm; (**g**) $h_1$—900 nm, $h_2$—65 nm; (**h**) $h_1$—1500 nm, $h_2$—20 nm; (**i**) $h_1$—1500 nm, $h_2$—30 nm; (**j**) $h_1$—2000 nm, $h_2$—30 nm; (**k**) $h_1$—300 nm, $h_2$—30 nm, $T$ = 300 °C.

Obviously, the adhesive forces of interaction between the silver layer and the monomer layer also contribute to the final polymer morphology. Depending on the polymer thickness and the amount of silver on the surface, the samples are characterized by different numbers of cracks. In general, the thicker the polymer and silver films, the more pronounced the cracks in the samples. Some samples were annealed at temperature $T = 300$ °C in air in order to study the effect of annealing on the surface morphology and on the nature of the enhancing properties. It can be seen (Figure 2c,k) that the globular Ag clusters coalesce into larger structures, and the ratio of adhesive and cohesive forces leads to the formation of deep and wide cracks in the sandwich structure. On the surface we see an enlargement of silver aggregates forming extensive islet agglomerations of about 1 μm in size, as shown in Figure 2k.

Figure 3 shows the AFM profilograms for various types of samples corresponding to four different groups according to the type of surface and the magnitude of the SERS signal enhancement.

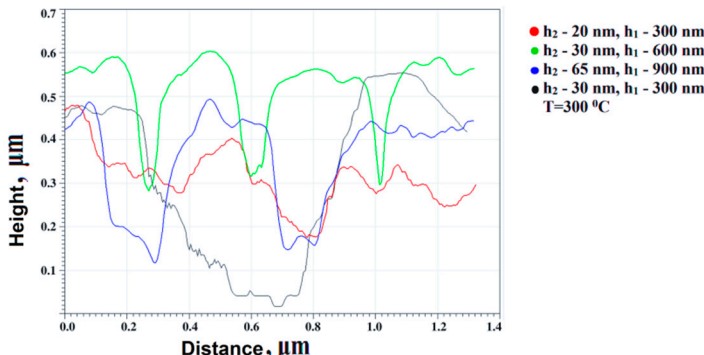

**Figure 3.** AFM profilograms: $h_1$—300 nm, $h_2$—20 nm (red); $h_1$—600 nm, $h_2$—30 nm (green); $h_1$—900 nm, $h_2$—65 nm (blue); $h_1$—300 nm, $h_2$—30 nm, $T = 300$ °C (black).

On the AFM profilograms it can be seen that the samples of the first group (Figure 3, red curve) without pronounced cracks on the surface (Figure 2b) had relief features with a height of about 150 nm. The next group of films (Figure 3, green curve) was characterized by regular cracks 200–250 nm deep and about 100 nm wide. For substrates with large polymer and silver thicknesses (Figure 3, blue curve), and for the annealed sample (Figure 3, black curve), large discontinuities were observed with an average depth of 600 nm and a midline width of 600 nm.

Table 2 shows the values of the film roughness parameters obtained from the Gwyddion software package. The mean-square roughness (*rms*), the height at three points ($R_{3z}$), the excess ($R_{ku}$), the mean-square waviness ($W_q$), and the average profile wavelength ($\lambda_a$) were calculated.

The *rms* remained almost constant for all films with PCPX thicknesses of 300 and 600 nm, except for a sample with thicknesses of $h_1 = 300$ nm, $h_2 = 30$ nm, for which *rms* was much larger. With an increase in the polymer layer thickness, *rms* decreases by about a factor of two. The $R_{3z}$ parameter increases significantly with an increase in the thickness of the upper Ag layer to 30 nm, but decreases noticeably with an Ag upper thickness of more than 50 nm and a polymer thickness of more than 900 nm. The parameter $R_{ku}$ decreased upon annealing samples and was characterized by increased values for thicker sandwich structures, starting from $h_1 = 900$ nm and $h_2 = 65$ nm. Thinner samples had $R_{ku}$ about one less, with the exception of the structure with parameters $h_1 = 600$ nm and $h_2 = 30$ nm. $W_q$ also decreased during annealing, while increasing significantly with increasing thickness of the silver layer to 30 nm with a polymer thickness of 300 nm. The parameter $\lambda_a$ was higher on average for samples with a polymer thickness of 900 nm or more. Thinner sandwich structures had a lower average profile wavelength.

**Table 2.** Morphology parameter.

| $h_1$, nm | $h_2$, nm | *rms*, nm | $R_{3z}$, nm | $R_{ku}$ | $W_q$, nm | $\lambda_a$, nm |
|---|---|---|---|---|---|---|
| 300 | 10 | 50 ± 7 | 229 ± 30 | 4.3 ± 1.1 | 48 ± 7 | 584 ± 76 |
| 300 | 20 | 49 ± 6 | 198 ± 22 | 2.9 ± 0.2 | 30 ± 5 | 513 ± 66 |
| 300 | 30 | 86 ± 13 | 383 ± 45 | 3.4 ± 0.8 | 90 ± 12 | 614 ± 75 |
| 600 | 30 | 50 ± 8 | 224 ± 29 | 4.1 ±1.1 | 35 ± 7 | 466 ± 54 |
| 900 | 30 | 73 ± 8 | 337 ± 35 | 3.1 ± 0.7 | 151 ± 45 | 800 ± 76 |
| 900 | 50 | 19 ± 4 | 104 ± 28 | 3.2 ± 0.7 | 73 ± 21 | 798 ± 68 |
| 900 | 65 | 44 ± 5 | 206 ± 36 | 4.4 ± 0.9 | 39 ± 18 | 619 ± 53 |
| 1500 | 20 | 21 ± 3 | 77 ± 31 | 4.1 ± 1.0 | 68 ± 23 | 916 ± 79 |
| 1500 | 30 | 25 ± 5 | 100 ± 41 | 4.0 ± 1.0 | 43 ± 15 | 953 ± 85 |
| 2000 | 30 | 24 ± 3 | 109 ± 45 | 4.8 ± 1.2 | 199 ± 43 | 614 ± 54 |
| 300 T = 300 °C | 30 | 51 ± 7 | 208 ± 19 | 2.7 ± 0.4 | 63 ± 13 | 517 ± 44 |

Note: The color indicates the groups of substrates as the signal decreases: red–orange–blue–green.

Thus, in each group of substrates, the roughness parameters of samples for which the thickness of the silver film was 30 nm were special. The maximum values of *rms* and $R_{3z}$ correspond to the sample with $h_1$ = 300 nm and $h_2$ = 30 nm; the highest values of the parameters $W_q$ and $R_{ku}$ correspond to the sample with $h_1$ = 2000 nm and $h_2$ = 30 nm; and the film with $h_1$ = 1500 nm and $h_2$ = 30 nm had the maximum average profile wavelength. Next, we present the correlations between the morphological parameters and the SERS signal.

*3.2. SERS Spectra*

The results of the DTNB SERS signal mapping over the surface are presented in Figure 4. Using these images, we can evaluate the uniformity of the signal distribution on the surface of the sandwich structure.

A control experiment was preliminarily carried out and the spectrum of a clean substrate was obtained. It is shown in Figure 4c in green. It is shown that its intensity is low and its own vibration bands do not coincide with the spectrum of the analyte DTNB.

Figure 4a shows an optical view of the sample's surface. The red square marks the scan area. Figure 4b shows the scan result in the form of the intensity distribution of the DTNB vibration band at 1338 cm$^{-1}$. Bright dots correspond to areas with the maximum signal. It can be seen that the signal is present everywhere, even in visually dark areas, but the distribution is heterogeneous, which may be due to the presence of optimal silver aggregates with the maximum enhancement effect. Figure 4c shows the characteristic DTNB spectrum, where the vibration bands according to [31] correspond to 1338 and 1561 cm$^{-1}$ arising from the asymmetric stretching mode of nitro groups $v_s$ (NO$_2$) and an aromatic ring stretching mode. The peaks at 1157 and 1064 cm$^{-1}$ were attributed to CH$_3$ rocking, C–N stretching, and C–N bending [32].

Figure 4c shows the spectra measured in the most SERS active region of the substrate (intensity 8000) and in the least active region (intensity 3000). The difference in intensities by several times shows that the substrate is not absolutely uniform. This is a consequence of the existence of silver aggregates of different sizes and shapes. However, the spectrum is present everywhere, which indicates that the substrate is sufficiently effective for routine use.

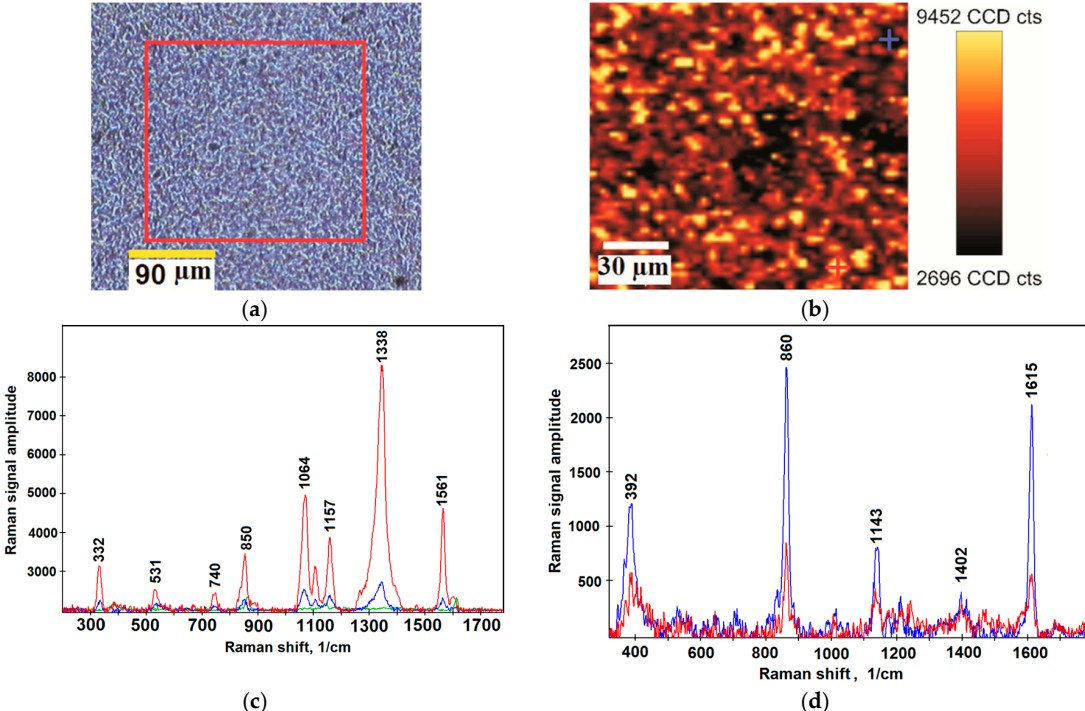

**Figure 4.** The results of the DTNB SERS signal investigation. (**a**) Optical image of the surface where the mapping area is shown by the red square; (**b**)mapping result of samples with $h_1$—300 nm, $h_2$—30 nm. The distribution of the amplitude of the main vibration band of the DTNB—1338 cm$^{-1}$ is presented in the form of a 2D map; (**c**) DTNB spectra at the brightest (red) and darkest (blue) scan points of the sample; and green—spectrum of clear substrate. The red cross in Figure 4b shows the place where at the spectrum with a higher amplitude was measured and the blue cross shows the place where at the spectrum with a lower amplitude was measured; (**d**) blue—spectrum of pure substrate, and red—spectrum after modification of the substrate with a solution of myoglobin.

Analysis of the measurement results was carried out by two methods. First, the dependencies of the DTNB spectrum amplitude on the thickness of silver and polymer were determined. Secondly, measurements of the integrated amplitude of the spectra from the entire scan area were analyzed. Based on these, the integral dependencies of the amplitude of the analyte vibrational band on the thickness of the silver layer in the sample were determined.

The dependencies of the amplitude of the SERS signal on the silver layer thickness and on the PCPX thickness were plotted in a two-dimensional representation (Figure 5). According to the dependence obtained, the signal maximum falls in the interval of polymer thickness between 300 and 600 nm with a silver layer thickness of 30 nm. With a further increase in the thickness of the silver layer, a decrease in the signal was observed, which can be explained by a significant change in the structure of the silver film on the surfaces of the samples, which enhanced the SERS signal. Significant deterioration of the signal was observed with an increase in the polymer thickness in the range of 800–2000 nm.

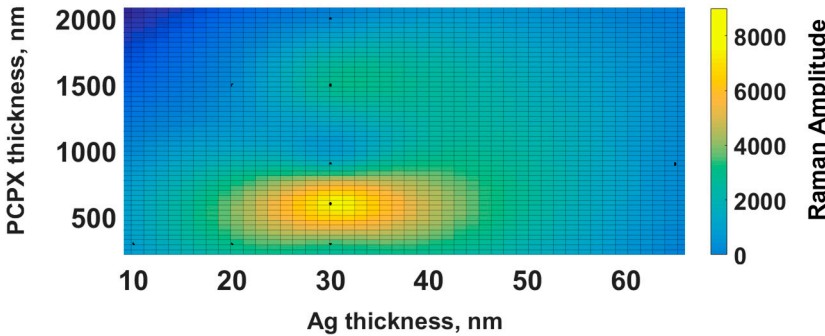

**Figure 5.** 2D representation of the dependence of the SERS signal on the silver layer thickness and the thickness of the PCPX layer on the substrate.

Figure 6 graphically presents the results for the dependence of the signal amplitude (Figure 6a) and the dependence of the integral signal (Figure 6b). The integral signal is the result of mathematical processing of the mapping results. This is the result of summing the amplitudes of the selected vibration band over the entire mapping area. In our case, the amplitudes of the vibration band at 1338 cm$^{-1}$ were summed up, since it was the highest for the DTNB. The use of the integrated signal technique made it possible to improve the statistical results of research, since large arrays of spectra were measured, and the influence of inhomogeneities of the studied medium decreased. The plots show a signal growth with increasing silver layer thickness and an increase in polymer thickness. Low confidence intervals were also obtained, which indicates the homogeneity of the surface of the substrate from the point of view of receiving a Raman signal. This favors the use of sandwich structures which are a silver film on a polymer as SERS active substrates.

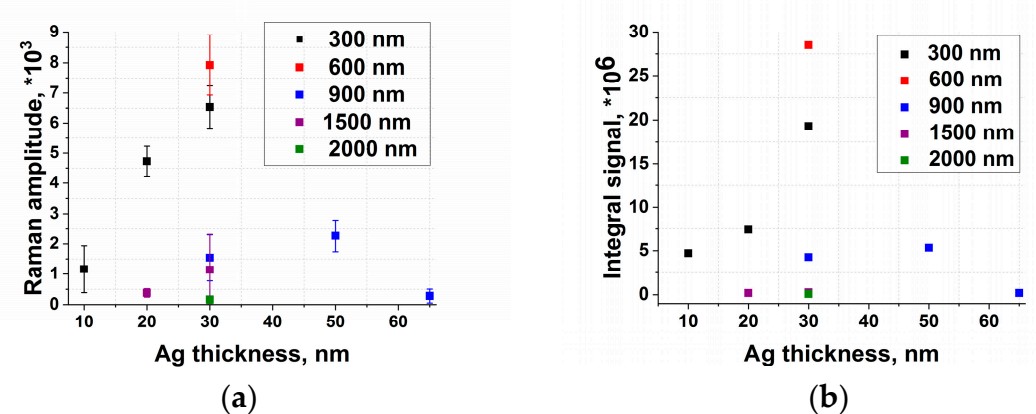

**Figure 6.** The study of Raman signals for samples of sandwich structures. The data are given for the amplitudes of the vibration band at 1338 cm$^{-1}$. (**a**) Dependence of the signal amplitude on the thickness of the silver layer; (**b**) dependence of the integral signal as a result of summing the amplitudes of the vibration band at 1338 cm$^{-1}$ over the entire mapping area on the thickness of the silver layer over an area of $150 \times 150$ μm$^2$.

It should be noted that the use of annealing showed a deterioration in the enhancing properties of the substrates until their complete disappearance, which was confirmed by the results of the morphological study. The restructuring of the structure upon annealing led to an increase in the intervals between the aggregates until the hot spots disappeared, and accordingly, the SERS signal disappeared.

When working with complex solutions, including high molecular weight and low molecular weight compounds in equal concentrations, the absence of SERS spectra for a high molecular weight

compound, for example, myoglobin and albumin proteins, was found. This suggests that the substrates presented are selective and can be used to separate mixtures of complex composition.

In Figure 4d, we present the spectrum of the substrate modified by protein in comparison with the spectrum of the pure substrate. As we can see, no new vibrational bands appeared. Myoglobin concentration was 0.2 g/L, similar to DTNB. This confirms our assumption about the selectivity of the presented substrate. We also saw a decrease in the intensity of the spectrum after applying myoglobin. The value of the spectrum intensity after treatment with a myoglobin solution changed from 2500 to 1000 for the vibration band at 860 cm$^{-1}$. The intensity of the pure substrate was higher than the intensity of the spectrum after DTNB treatment, since DTNB was chemically bound to silver nanoparticles and aggregated and screened the possibility of enhancing the spectrum of the substrate itself. Additionally, this may have been due to the shielding of the spectrum by an optically transparent material due to additional light scattering.

Figure 7 shows the correlation of the SERS signal with the basic parametric characteristics of the surface.

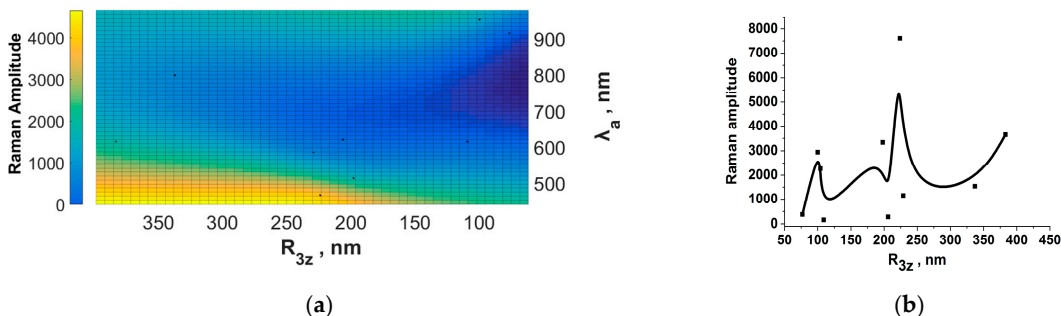

(**a**)    (**b**)

**Figure 7.** Dependence of signal amplitude on roughness parameters: (**a**) height at three points ($R_{3z}$) and average profile wavelength ($\lambda_a$); (**b**) SERS signal amplitude vs. height at three points $R_{3z}$.

From the analysis of these results it follows that the maximum value of the Raman signal was shown by samples with an average profile wavelength of 466–614 nm. These values correspond to the disc sizes of silver films obtained in [33]. In this study, systems with silver aggregates were created, which can be represented by 646 nm microdiscs, between which an increase in the local electric field was observed. We observed these dimensional features just for a group of films with morphological indices characteristic of $h_1$ 600 nm $h_2$ 30 nm

It has been revealed that the average profile wavelength and height at three points play the largest roles from the point of view of plasmon resonance realization. The optimal morphology was characterized by the closest similarity of the linear dimensions of silver agglomerates formed on the surface to the wavelength of the exciting radiation. In this case, the minimum distance between them (from units to tens of nm) was observed, which corresponds to the conditions of the most efficient excitation of plasmon resonance.

We assume that it was the combination of a polymer thicknesses of 600 nm and silver of 30 nm as a result of the complex processes of heat transfer and mass transfer during polymer polymerization that led to the most favorable morphological state of the surface, which exhibited optimal plasmon properties for amplification at a wavelength of 785 nm. We used ellipsometry to show that at these thicknesses, the position of the plasmon resonance coincides with the excitation wavelength of 785 nm.

*3.3. Ellipsometry*

The ellipsometric parameters Ψ and Δ were measured in the wavelength range of 380–980 nm according to the basic ellipsometry equation (Equation (1)):

$$\rho = \frac{R_p}{R_s} = \tan(\psi)e^{-i\Delta} \tag{1}$$

where $R_{\text{p}}$ is the amplitude after reflection of the p-component and $R_{\text{s}}$ is the amplitude after reflection of the s-component [34,35]. The samples were illuminated by a parallel beam of polarized light using the geometry of oblique incidence at an angle $\theta = 20°$.

To model the results we used the Maxwell–Garnett effective medium equation, which allows the replacement of the heterogeneous film with an equivalent homogeneous medium with an effective dielectric susceptibility defined by Equation (2), where $\varepsilon$ is the effective dielectric constant of the system, $\varepsilon_\alpha$ is the dielectric constant of inhomogeneities (silver) with the volume content $f_\alpha$, and $\varepsilon_\beta$ is the dielectric constant of the matrix (PCPX).

$$\frac{\varepsilon - \varepsilon_\beta}{\varepsilon + 2\varepsilon_\beta} = f_\alpha \frac{\varepsilon_\alpha - \varepsilon_\beta}{\varepsilon_\alpha + 2\varepsilon_\beta} \tag{2}$$

In the Spectr software a substrate model was constructed consisting of a composite layer with thicknesses of 10, 20, and 30 nm, corresponding to the silver thicknesses on a pure PCPC layer with thicknesses 300 nm, 600 nm, and above, corresponding to the polymer thickness, and in turn, located on an optically opaque silver layer 100 nm thick. In this geometry, the calculations cannot take into account the polycor substrate. For calculations, we used the optical characteristics of silver from [36] and the optical characteristics of PCPX from [37]. The concentration of silver in the nanocomposite and the thickness of the composite layer were simulated. As a result of the calculation, the best parameters were selected by minimizing the MSE (mean-square error) calculation.

Thus, the theoretical dependencies of the ellipsometric angles $\Psi$ and $\Delta$ on the wavelength were calculated. Figure 8 presents graphs illustrating a comparison of experimental and theoretical dependencies of the ellipsometric angles $\Psi$ and $\Delta$ on wavelength for samples with the following parameters: $h_1$—300 nm, $h_2$—30 nm (Figure 8a,b); $h_1$—600 nm, $h_2$—30 nm (Figure 8c,d); $h_1$—300 nm, $h_2$—30 nm, $T = 300°C$ (Figure 8e,f).

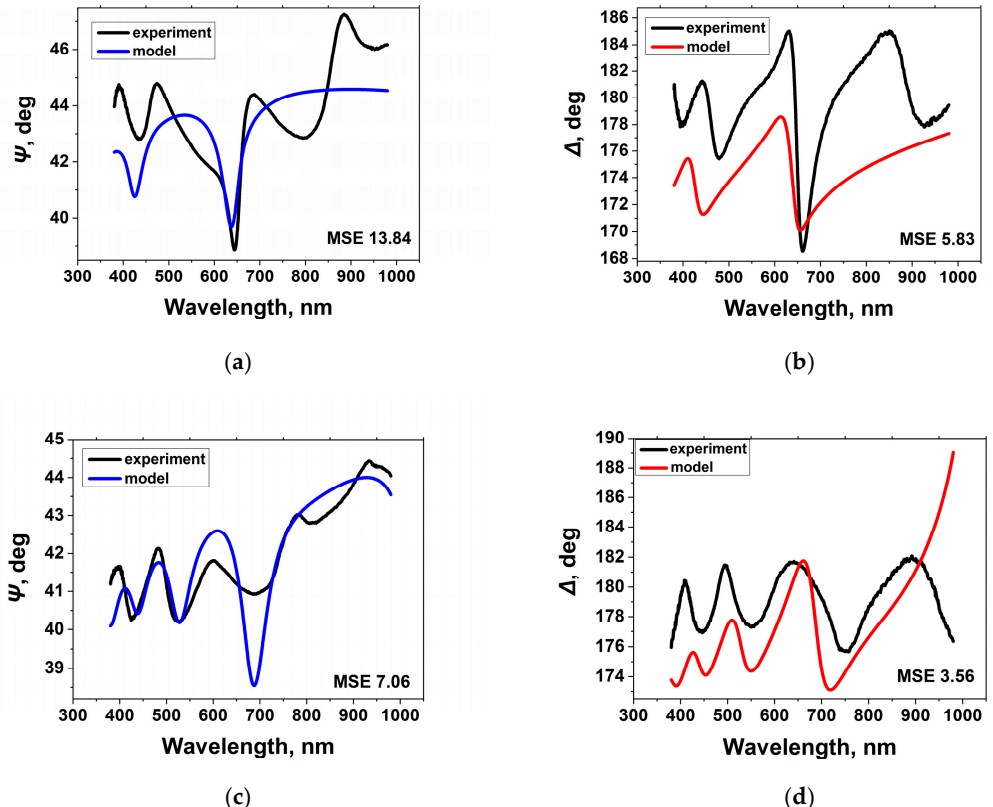

**Figure 8.** *Cont.*

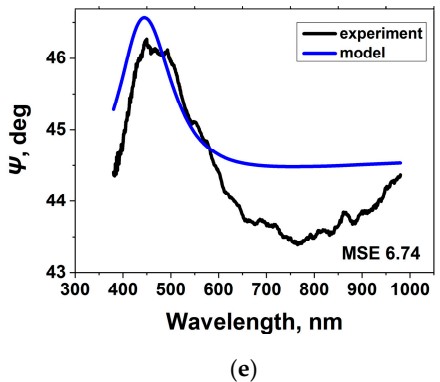

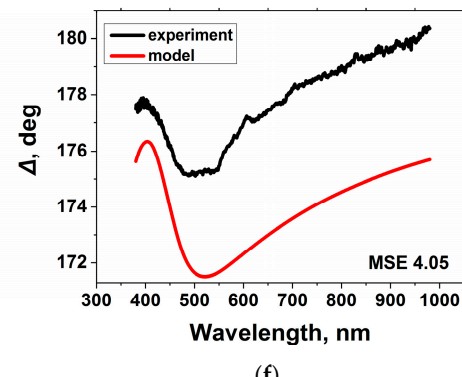

(**e**)　　　　　　　　　　　　　　　　　　　　　　　　　　　　　(**f**)

**Figure 8.** Comparison of the experimental (black) and theoretical dependencies of the ellipsometric angles Ψ (blue) and Δ (red) on the wavelength at an angle of incidence of 20° for the samples: (**a**,**b**): $h_1$—300 nm, $h_2$—30 nm; (**c**,**d**): $h_1$—600 nm, $h_2$—30 nm; (**e**,**f**): $h_1$—300 nm, $h_2$—30 nm, $T$ = 300 °C (where $h_1$—polymer thickness, $h_2$—silver thickness, MSE—mean-square error).

On the spectra in Figure 8a,b plasmon resonance is present at wavelengths of about 650 and 900 nm, which do not coincide exactly with the SERS excitation wavelength, which in our case was 785 nm. Therefore, although high-amplitude Raman spectra were obtained for this sample, they were not at a maximum. In the Ψ and Δ spectra of the sample with parameters $h_1$—600 nm, $h_2$—30 nm, the resonance was located near the wavelength of the exciting radiation, due to which this sample exhibited the maximum spectrum gain compared to rest of the samples. There was no resonance in the Ψ and Δ graphs of the annealed sandwich structure, which is consistent with the absence of the SERS signal and is explained by the combination of silver aggregates into large agglomerates on the surface of the substrate, leading to the destruction of the plasmon structure.

Some discrepancy between the experimental and theoretical data of Ψ and Δ spectra in Figure 8 is obvious, which is explained by the fact that in the described model of the sandwich structure, the upper effective layer consisting of silver and polymer is not ideally dispersed relative to silver nanoparticles. However, even with the observed mismatch, we showed the existence of plasmonic properties precisely at the excitation wavelengths. It is a qualitative assessment in this case that is sufficient to substantiate the effectiveness of sandwich structures.

We did not present the data on the ellipsometric parameters of the remaining samples because of their uniformity; however, they showed the absence of a noticeable plasmon resonance at the wavelength of the exciting radiation, which correlates with lower values of the amplitude of the SERS signals.

## 4. Conclusions

Using gas-phase cryochemical synthesis and thermal sputtering in vacuum, SERS-active sandwich structures of Ag–PCPX–Ag were obtained. In the fabrication process of these sandwich structures, the upper silver film was formed with a complex island topology with submicron sizes.

SERS spectra were obtained from an analyte chemically adsorbed on the films' surfaces of various thicknesses, demonstrating a significant enhancement of the spectrum (up to $10^4$) compared with the control sample. The dependence of the gain on the silver concentration is characterized by a maximum with a polymer layer 600 nm thick and a 30 nm thick upper silver layer. A protein-selective enhancement of the SERS spectra amplitudes using a low molecular weight compound was observed.

The maximum value of the SERS signal was shown by samples with an average profile wavelength of 466–614 nm. According to the results of AFM studies, it was revealed that the largest roles from the point of view of plasmon resonance implementation were played by the average profile wavelength and the average relief height at three points.

The optimal morphology was characterized by the closest proximity of the linear dimensions of the flat silver fragments formed to the wavelength of the exciting radiation. In this case, the minimum distance between them (from units to tens of nm) was observed, which corresponds to the conditions of the most efficient excitation of plasmon resonance.

Based on the proposed model and ellipsometric measurements, by minimizing the error expressed as the minimum standard deviation, the theoretical dependencies of the ellipsometric angles $\Psi$ and $\Delta$ on the wavelength were calculated. The ellipsometric parameters of most of the samples did not show the existence of plasmon resonance at the wavelength of the exciting radiation. In the $\Psi$ and $\Delta$ spectra with 30 nm Ag layer thicknesses at 600 nm PCPX, the resonance is near the wavelength of the exciting radiation, which indicates the existence of plasmon resonance, which manifests as a maximum in the SERS spectra.

The films obtained are promising for the selective registration of low molecular weight organic compounds in solutions containing proteins by the method of recording SERS spectra.

**Author Contributions:** Conceptualization, I.B., I.R. and A.G. (Alexey Gusev); methodology, A.V.; software, A.G. (Alexey Gusev); validation, I.B., A.G. (Aliia Gainutdinova) and A.L.; formal analysis, M.S. and S.C.; investigation, A.M., I.B., A.G. (Alexey Gusev) and A.G. (Aliia Gainutdinova); resources, K.M.; data curation, A.L. and I.R.; writing—original draft preparation, A.G. (Alexey Gusev); writing—review and editing, M.S. and I.R.; visualization, A.G. (Aliia Gainutdinova); supervision, A.L.; project administration, I.R. All authors have read and agreed to the published version of the manuscript.

**Funding:** This research was funded by RFBR, project number 20-08-00632 A.

**Acknowledgments:** Samples were made with ITAE RAS laboratory equipment.

**Conflicts of Interest:** The authors declare no conflict of interest. The funders had no role in the design of the study; in the collection, analyses, or interpretation of data; in the writing of the manuscript, or in the decision to publish the results.

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
