# Peer review of "Detection of Organic Substances by a SERS Method Using a Special Ag-Poly(Chloro-P-Xylylene)-Ag Sandwich Substrate"

_coatings, doi:10.3390/coatings10080799_

Round 1
Reviewer 1 Report
The results are promising for the selective registration of low molecular weight organic compounds in solutions containing proteins by the method of recording SERS spectra. The followings are my comments for this manuscript with minor revision.
- The surface roughness of the polycrystalline Al2O3 (polycor) substrate has to be described in Figure 1.
- The dependence of the gain on the silver concentration is characterized by a maximum with a polymer layer 600 nm thick and a 30 nm thick upper silver layer. More explanation about both thickness effects on the SERS should be provided.
- The reason for choosing the PCPX in the Ag-PCPX-Ag structure for SERS is not clear. The authors have to compare with the other similar SERS structures and make possible benchmark.
Author Response
Dear Reviewer,
We sincerely thank you for your questions that will help make our article better. We have done a thorough job to answer your comments as best as possible. The article has been modified accordingly according to your questions. The new version of the article is also attached.
- The surface roughness of the polycrystalline Al2O3 (polycor) substrate has to be described in Figure 1.
Answer.
We have presented an AFM image of polycrystalline Al2O3 in Figure 1 (b) with an indication of its roughness and described the result obtained in lines №№ 177 – 179 of main paper text. It should be noted that its roughness value is much less than the obtained roughness for the final SERS structures considered by us in this work. Therefore, we can say that the material used as a base substrate does not significantly affect the final morphological properties of the sandwich structures. Its choice is due to the good adhesion between it and silver, which makes the SERS sandwich structures stable when working with solutions.
- The dependence of the gain on the silver concentration is characterized by a maximum with a polymer layer 600 nm thick and a 30 nm thick upper silver layer. More explanation about both thickness effects on the SERS should be provided.
Answer.
We assume that it is the combination of polymer thicknesses of 600 nm and silver of 30 nm as a result of complex processes of heat transfer and mass transfer during polymer layer polymerization that led to the most favorable morphological state of the surface, which exhibits optimal plasmon properties for SERS enhancement at a wavelength of 785 nm. In this work, we used ellipsometry to show that at these thicknesses, the position of the plasmon resonance coincides with the excitation wavelength of 785 nm.
In the paper text in lines № № 357 – 362 and 369 -- 374 we describe our assumptions about the effect of thickness. We assume on the basis of the AFM studies that at a given combination of thicknesses, as a result of the polymerization and dispersion of silver nanoparticles on the surface, an optimal morphology is formed. This occurs as a result of the processes of self-formation and self-ordering, which is an experimental fact. In an earlier study in [33], nanostructures were described, in which, as a result of theoretical modeling, the conditions for the realization of an amplifying electromagnetic field were shown. The presented geometric model is similar to those considered in our work in terms of geometric dimensions.
- The reason for choosing the PCPX in the Ag-PCPX-Ag structure for SERS is not clear. The authors have to compare with the other similar SERS structures and make possible benchmark.
Answer.
The group of polymers of poly (paraxylylene) derivatives is unique, as it has the property of polymerizing from the gas phase in the process of cryosynthesis in the form of thin films on substrates. This allows them to be simultaneously applied with streams of atomic silver nanoparticles. This makes it possible to create truly nanoplasmonic coatings that are conformal to the substrate surface and chemically resistant, which was shown in our works [1, 14]. Chlorine-substituted polymer is the most thermally and chemically stable polymer from the group of poly-(para-xylylene). This removes the risks when working with organic solvents. This is particularly important in experiments with biological samples.
In the main text of our paper in lines №№ 117 - 135 we have supplemented the text about the relevance of our work and the choice of this polymer. It should be noted that the process we propose is closed and allows the formation of a finished SERS coating completely inside the installation without the use of additional technological processes, solvents, and aggregators. This makes the process simple and cheap and greatly improves reproducibility.
With kinde regards,
Dr. Alexey Gusev
Reviewer 2 Report
The authors have developed thin-film sandwich structures in which, when used as sensitive elements for detecting organic compounds at low concentrations, high-amplitude spectra of surface enhanced Raman scattering (SERS) are observed. The work needs major revision.
- The novelty of the work must be highlighted in the introduction section.
- The ellipsometry section must be improved : For example the difference between generated and experimental data should be improved. Could the authors plot the difference between generated and experimental data?
- Moreover, the authors could plot some literature form the data and made a comparison between their data and literature data.
- The authors should add a discussion about previous works on sandwich structure:
See for example:
Fabrication of Au/graphene oxide/Ag sandwich structure thin film and its tunable energetics and tailorable optical properties, AIMS Mater. Sci. 4 (n.d.) 223–230. https://doi.org/http://dx.doi.org/10.3934/matersci.2017.1.223.
Sandwich-structured Ag/graphene/Au hybrid for surface-enhanced Raman scattering, Electrochim. Acta. 119 (2014) 43–48.
Micro-Raman investigation of Ag/Graphene oxide/Au sandwich structure, Mater. Res. Express. (2019). https://doi.org/10.1088/2053-1591/ab11f8.
Author Response
Dear Reviewer,
we sincerely thank you for your questions that will help make our article better. We have done a thorough job to answer your comments as best as possible. The article has been modified according to your comments. The new version of the article is also attached.
- The novelty of the work must be highlighted in the introduction section.
Answer.
In the introduction of the main text of the article in lines №№ 55 - 135 we provided additional data on this issue. Indeed, our initial introduction was incomplete. We note additionally that the global novelty of the work lies in the fact that we were the first to use the technology of cryosynthesis in vacuum to create SERS of an active medium based on sandwich structures. We used such a combination of the refractive and extinction coefficients of the polymer and metal to create a medium with a plasmon resonance position at a wavelength close to 785 nm, which is used in this work to record the SERS signal. We used a unique polymer capable of polymerizing from the gas phase without the participation of solvents and other accompanying reagents. In a vacuum chamber at a sufficiently high vacuum, as a result of the process of self-formation and self-ordering, simultaneously with the process of polymerization, an environment is formed that implements the SERS effect. The polymerizing polymer controls the morphology of the plasmonic active layer in a vacuum in a clean environment, while it also acts as a spacer to form an optically active medium.
- The ellipsometry section must be improved : For example the difference between generated and experimental data should be improved. Could the authors plot the difference between generated and experimental data?
Answer.
We used ellipsometry in our study to qualitatively prove the existence of plasmonic properties in our nanostructures. The Maxwell-Garnett effective medium equation used is suitable for evenly distributed nanoparticles in the matrix. In our case, only a qualitative assessment of the existence of plasmonic properties is really possible. Incomplete coincidence of the experiment and the calculation suggests that our environment is not completely homogeneous. But this does not affect the implementation of the SERS effect as a whole. We can use such an estimate, since our goal is to assess the presence or absence of plasmonic properties that show the suitability of the optical system for realizing the SERS effect. But we do not aim to accurately determine the effective dielectric constant. Since the working layer is morphologically heterogeneous and difficult for such assessments. In the main text of the article in lines №№ 415 - 420, we also substantiate this answer.
- Moreover, the authors could plot some literature form the data and made a comparison between their data and literature data.
Answer.
According to your comment, we have supplemented our literature review and significantly expanded the introduction, starting from line № 55 to line № 135. We indicated that there are a number of modern works on the creation of active SERS structures, each of which has its own advantages. We considered a number of works on the creation of SERS active structures based on sandwich structures. Conducted an analysis of the benefits and highlighted areas in which the work we submit is relevant.
- The authors should add a discussion about previous works on sandwich structure:
See for example:
Fabrication of Au/graphene oxide/Ag sandwich structure thin film and its tunable energetics and tailorable optical properties, AIMS Mater. Sci. 4 (n.d.) 223–230. https://doi.org/http://dx.doi.org/10.3934/matersci.2017.1.223.
Sandwich-structured Ag/graphene/Au hybrid for surface-enhanced Raman scattering, Electrochim. Acta. 119 (2014) 43–48.
Micro-Raman investigation of Ag/Graphene oxide/Au sandwich structure, Mater. Res. Express. (2019). https://doi.org/10.1088/2053-1591/ab11f8.
Answer.
We answered this question together with the previous question, in which we used the works you kindly indicated to analyze the literature in conjunction with other contemporary works on this topic.
With kinde regards,
Dr. Alexey Gusev
Reviewer 3 Report
The authors describe a SERS substrate based on a sandwich structure of silver-polymer-silver. The paper is overall well written and provides sufficient detail on methodology and physical characterization. The simulations for plasmonic excitation strengthen the paper.
However, when it comes to the characterization of the SERS signals, the paper must be improved. The authors claim “detection of organic substances” but only demonstrate data from DTNB, with no information on concentration, no controls (substrate only), and no reference spectra (DTNB Raman). This information must be included.
Major concerns
The cited literature is dominated by older sources, which although historically important, do not convey the state of the art. The authors should include references to some more recent work on SERS substrates, e.g. from the Moskovits group.
In Figure 4, panel (b) shows that the intensity of the highest peak is >9,000 counts. Panel (c) shows the highest peak at ~5,000 counts. Why the discrepancy? Is it due to the baseline? The authors should make this consistent.
Also, please provide a spectrum of the substrate without analyte (DTNB) for reference.
In lines 240 to 243, in the abstract, and elsewhere, the authors compare signals from chemicals of low MW vs proteins. These data are not shown in the manuscript. Either include these measurements and appropriate controls or remove this claim throughout.
Minor comments
In the first line of the abstract, it is stated that Raman spectroscopy has extremely high sensitivity. This is false. The authors should clarify that they refer to SERS and not to other types of Raman spectroscopy.
Line 31, 104 not 104.
Line 64, is it truly 104? Now I’m doubting myself.
In figure 3, the plot is very confusing. Consider changing the axis labels to ‘length’ and ‘height’ or similar. Or clarify if this means something else.
Figure 4. Please add more information in the figure caption, especially regarding panel (b).
Also, can the points where the spectra were taken be indicated in the sample image?
Figures 5 and 7. The 3D representation is not clear and it is not necessary. Please show these graphs as 2D maps.
Figure 6, what is the 'integral signal'? Please explain this in the caption and the main text.
Author Response
Dear Reviewer,
we sincerely thank you for your questions that will help make our article better. We have done a thorough job to answer your comments as best as possible. The article has been modified according to your comments.
The new version of the article is also attached.
Major concerns
- The cited literature is dominated by older sources, which although historically important, do not convey the state of the art. The authors should include references to some more recent work on SERS substrates, e.g. from the Moskovits group.
Answer.
We have significantly supplemented and expanded our literary review and significantly expanded the introduction section with a review of the current state of the SERS research, starting with lines №№ 55 - 135. We used the work of the kindly indicated group of Moskovits and other scientific groups. We also analyzed the state of the art for sandwich structures research.
- In Figure 4, panel (b) shows that the intensity of the highest peak is >9,000 counts. Panel (c) shows the highest peak at ~5,000 counts. Why the discrepancy? Is it due to the baseline? The authors should make this consistent.
Answer.
Figure 4 (b) represents the distribution of the band intensity at 1338 cm-1 over the scan area. This distribution was obtained from mapping. The mapping mode allows you to build maps of the intensity distribution of the selected vibration band. In this case, the vibration band at 1338 cm-1 was chosen, since it is the most intense for the DTNB. A value >9000 is the maximum intensity. In Figure 4 (b), we show that we observe a change in intensities over the scan area from values >9000 to >2000. That is, we always have a signal, but its amplitude can vary over the scanning area. We marked the places where the spectra were measured with a blue and red cross, which corresponds to the maximum and minimum spectrum. We subtract the baseline from the entire array of measurements. To avoid confusion with the values, we have removed the intensity values in Figure 4 (c) and show qualitatively that the amplitudes of the spectra can be different and depend on the measurement site. Obviously, there was a misprint earlier, and we presented the spectra from the intermediate sections of the intensity map.
- Also, please provide a spectrum of the substrate without analyte (DTNB) for reference.
Answer.
In the text of the article, in Figure 4 (c) with green color and in Figure 4 (d) in red, the spectrum of the substrate without analytes is shown and its description is given in lines № 286 - 288. It is shown that the vibrational bands of the substrate do not coincide with the spectrum of the DTNB.
- In lines 240 to 243, in the abstract, and elsewhere, the authors compare signals from chemicals of low MW vs proteins. These data are not shown in the manuscript. Either include these measurements and appropriate controls or remove this claim throughout.
Answer.
We presented data on our studies of the myoglobin spectrum in Figure 4 с. And provided additional explanations in the text on these studies in lines №№ 347 - 352. We show that in work with myoglobin, we do not observe its spectrum. We present the spectra of the substrate without myoglobin and after substrate treatment with myoglobin. It can be seen from the spectra that there are no differences in the spectra. There is a change in the amplitude after the deposition of the analyte, which may be due to the shielding of the substrate by an optically scattering dielectric material, which is myoglobin.
Minor comments
- In the first line of the abstract, it is stated that Raman spectroscopy has extremely high sensitivity. This is false. The authors should clarify that they refer to SERS and not to other types of Raman spectroscopy.
Answer.
Yes, indeed, we formulated it inaccurately. And we fixed the first line in the abstract
- Line 31, 104 not 104.
Answer.
Sorry, there was a typo while loading the text and the value must be 10 ^ 4, not 104
- Line 64, is it truly 104? Now I’m doubting myself.
Answer.
Sorry, there was a typo while loading the text and the value must be 10 ^ 4, not 104
- In figure 3, the plot is very confusing. Consider changing the axis labels to ‘length’ and ‘height’ or similar. Or clarify if this means something else.
Answer.
Yes, in Figure 3, the axes in the previously presented version look uninformative. We have changed: X-axis - distance, microns; Y-axis - height, microns.
- Figure 4. Please add more information in the figure caption, especially regarding panel (b).
Answer.
We have expanded the description to figure 4. Including the text in lines №№ 286 – 288 and 322 - 328
- Also, can the points where the spectra were taken be indicated in the sample image?
Answer.
We have shown with red and blue crosses the places where the spectrum was measured.
- Figures 5 and 7. The 3D representation is not clear and it is not necessary. Please show these graphs as 2D maps.
Answer.
We have completed these Figures 5 and 7 as 2D maps
- Figure 6, what is the 'integral signal'? Please explain this in the caption and the main text.
Answer.
The integral signal is the result of mathematical processing of the mapping results. This is the result of summing the amplitudes of the selected DTNB vibration band over the entire mapping area. In our case, the summation of the amplitudes (or intensities) of the vibration band at 1338 cm-1 was carried out, since it is the highest amplitude for the DTNB. Changes were made to the main text in lines №№ 322 – 328 and in the figure caption.
With kinde regards,
Dr. Alexey Gusev
Round 2
Reviewer 2 Report
I recommend the publication of this article.
Author Response
Dear reviewer,
thank you for working with our article.
We have attached a version with minor text changes.
Best wishes, Alexey Gusev
Reviewer 3 Report
The manuscript is improved on many accounts.
I only have a problem with the decision of the authors to remove the units from Figure 4c and 4d "in order to avoid confusion". Unfortunately our field suffers from lack of quantitation, so we should avoid such bad practice if we want to help our colleagues.
I only request that the authors add the units back to their y-axes, and if there is a discrepancy due to baseline, or another form of spectral processing, then clarify it in the text.
Author Response
Dear reviewer,
according to your comment, Figures 4c and 4d have been changed. The Y scale has been restored. In the text in the lines №№ 307 – 311 and 356 - 360, the corresponding explanations were described.
Best regards,
Gusev Alexey